# NPCS: Native Provenance Computation for SPARQL

## ABSTRACT

The popularity of Knowledge Graphs (KGs) both in industry and academia owes credit to their flexible data model, which is suitable for data integration from multiple sources. Several KG-based applications such as trust assessment or view maintenance on dynamic data rely on the ability to compute provenance explanations for query results. The how-provenance of a query result is an expression that encodes the records (triples or facts) that explain its inclusion in the result set. This article proposes NPCS, a Native Provenance Computation approach for SPARQL queries. NPCS annotates query results with their how-provenance. By building upon spm-provenance semirings, NPCS supports both monotonic and non-monotonic SPARQL queries. Thanks to its reliance on query rewriting techniques, the approach is directly applicable to already deployed SPARQL engines using different reification schemes – including RDF*. Our experimental evaluation on two popular SPARQL engines (GraphDB and Stardog) shows that our novel query rewriting brings a significant runtime improvement over existing query rewriting solutions, scaling to RDF graphs with billions of triples.

## CCS CONCEPTS

• **Information systems** → **World Wide Web**; *Database query processing*.

### ACM Reference Format:
Anonymous Author(s). 2018. NPCS: Native Provenance Computation for SPARQL. In *Proceedings of The 38$^{th}$ Annual AAAI Conference on Artificial Intelligence (AAAI '24)*. ACM, New York, NY, USA, 9 pages. https://doi.org/XXXXXXX.XXXXXXX

## 1 INTRODUCTION

Thanks to the continuous advances in Web information extraction and knowledge graph construction, the Web nowadays enjoys from a plethora of machine-readable data, structured in large RDF knowledge graphs (KGs). These KGs, often queried via SPARQL endpoints, allow computers to "understand" the real world. They do so by encoding knowledge as collections of triples or statements $(s, p, o)$ with subject $s$, predicate $p$, and object $o$, e.g., (*UK, capital, London*). A triple $(s, p, o)$ can also be seen as a directed $p$-labeled edge from node $s$ to node $o$. This data model serves as foundation for multiple applications such as question answering, Web search, and smart assistants.

Given the heterogeneity of data sources that contribute to modern KGs, the problem of identifying the *provenance* of query results is central for many KG-based tasks. The provenance of a query result is an expression that encodes the lineage of data transformations and statements that contributed to that result. Provenance is of great value for KG providers because it streamlines maintenance tasks such as source selection and view maintenance. For data consumers, query provenance serves as an explanation for answers. This can be pivotal in use cases that need to assess data reliability, or manage access control, trustworthiness, and data quality.

Among the existing formalisms to model provenance, *how-provenance* is the most expressive [13] . In this model, the provenance of a query result is an algebraic expression in a *provenance semiring*. Consider, for instance the following KG,

$$\{\, u_1 \colon (UK, capital, London), u_2 \colon (Belfast, in, UK), u_3 \colon (Belfast, a, City) \,\},$$

and the SPARQL query

SELECT $?x$ WHERE $\{\ \{UK\ capital\ ?x\ \}\ $ UNION $\{?x\ in\ UK; a\ City\}\ \}.$

The solutions to this query are *Belfast* and *London*. How-provenance explains the presence of *London* in the result set with the polynomial expression $u_1 \oplus (u_2 \otimes u_3)$. This polynomial tells us that there are two way to get London as a solution: either via $u_1$, or via the conjunction of $u_2$ and $u_3$. There exist different algebraic structures for provenance in the literature [7, 10, 13], but this paper focuses on spm-semirings [10], because they are designed for the semantics of SPARQL, including its non-monotonic fragment. We highlight that query provenance assumes the availability of identifiers for triples, in order words, it assumes that the KG has been reified using some scheme. Examples of reification schemes are RDF* and named graphs.

There are essentially two main strategies to compute how-provenance annotations for query results. Methods such as TripleProv [17], opt for customized engines designed to compute provenance along query evaluation. Since provenance support is embedded in the engine, such solutions allow for advanced optimizations. On the downside, customized engines are not applicable to already deployed SPARQL endpoints in the Web. The other alternative is query rewriting [15]. By this logic, SPARQL queries are rewritten so that the new query retrieves both the query solutions and the polynomials that describe their provenance, potentially with some post-processing. While rewriting the query induces a runtime overhead, this design provides the flexibility to be applied to any SPARQL endpoint on the Web.

With this in mind we propose NPCS, a new query rewriting method for how-provenance computation in RDF/SPARQL engines. Unlike previous approaches [15], NPCS does not require any post-processing to compute query solutions with how-provenance annotations. This makes NPCS the first fully native SPARQL solution for how-provenance. Moreover, NPCS supports different data reification schemes, including RDF*. Our experimental evaluation on both synthetic and real data suggest that our fully native SPARQL rewriting (a) incurs a reasonable runtime overhead, and (b) it is consistently faster than SPARQLprov [15], the state of the art in how-provenance in SPARQL.

 

The rest of the paper is structured as follows. In Sections 2 and 3, we provide the basic background for this work in terms of related work and preliminaries respectively. In Section 4 we describe in detail our rewriting method, whereas Section 5 provides a comprehensive evaluation of the performance of our approach. We conclude in Section 6.

## 2 RELATED WORK

We survey the literature on provenance for query results along two axis: provenance models (Section 2.1), and provenance support for RDF/SPARQL engines (Section 2.2).

### 2.1 Semirings and Provenance Models

Semirings were first used to model query provenance in the ground-breaking work by Green et al. [13]. This work proposed *commutative semirings* to annotate query results for selection, projection, join, and union queries for Datalog and the positive fragment of relational algebra. Commutative semirings cannot model provenance for non-monotonic operators such as the left-outer join and the difference [13], hence the algebraic structures were expanded to include a monus operator that accounts for the relational difference [9]. Commutative semirings and their extensions model provenance as polynomial expressions. These expressions, called *how-provenance*, encode both the sources and the data transformations required to obtain (and sometimes exclude) a particular query answer. How-provenance is more expressive than other provenance models such as lineage [6] or why-provenance [5].

Damasio et al., [7] showed that by rewriting SPARQL queries into relational algebra, it is possible to provide provenance annotations for SPARQL queries using *m-semirings*. These, however, can yield very long and complex provenance expressions [10], as shown by Geerts and his colleagues. They therefore developed the *spm-semirings* formalism (spm stands for SPARQL Minus) to overcome these limitations. Spm-semirings guarantee more compact explanations and offer native support for non-monotonic SPARQL operators such as OPTIONAL and MINUS. Since how-provenance polynomials are abstract annotations, they are useful to a handful of metadata management applications [8, 14] via the notion of commutation with homomorphisms.

### 2.2 Provenance-supported SPARQL engines

Wylot et al. [17] introduced TripleProv, a system to compute how-provenance annotations in the commutative semiring model. This approach supports queries with basic graph patterns, union, and the OPTIONAL operator. Because of its reliance on commutative semirings, TripleProv cannot guarantee commutation with homomorphisms for queries involving the non-monotonic OPTIONAL operator. Additionally, TripleProv uses a customized engine that organizes data into molecules—sort of indexes for star patterns. As a result, TripleProv cannot be used on already deployed SPARQL engines. This is why our approach resorts to query rewriting for how-provenance computation.

But approaches based on query rewriting are not rare at all. Perm [12] and GProM [3] are two examples. Such approaches are tailored for relational databases, hence they are not applicable to SPARQL queries out of the box. Similarly to TripleProv, none

of these methods can properly support non-monotonic SPARQL queries, because Perm is based on the lineage model, and GProM relies on commutative semirings.

While the work of Geerts et al [10] was the first to study provenance for the non-monotonic fragment of SPARQL, the first concrete method to compute how-provenance under the spm-semiring formalism was proposed by Hernandez et al. [15]. They introduced SPARQLprov, a method based on query rewriting that can annotate query results with how-provenance polynomials for both monotonic and non-monotonic queries. Contrary to NPCS, SPARQLprov is not a 100% SPARQL solution, because it relies on a subsequent decoding phase to compute the final provenance annotations from the results of the rewritten query. As our experimental evaluation shows, this decoding phase can incur prohibitive runtime overheads for non-selective queries.

## 3 PRELIMINARIES

### 3.1 RDF* and SPARQL*

We assume the existence of three (pairwise disjoint) countably infinite sets: the set of IRIs $I$, the set of blank nodes $B$, and the set of literals $L$. An RDF triple $t = (s, p, o) \in T$, where $T = (I \cup B) \times I \times (I \cup B \cup L)$, is a statement that consists of a *subject* $s$, a *predicate* $p$, and an *object* $o$. A collection of RDF triples $\Gamma$ is called an RDF graph. The RDF* data model extends RDF by allowing arbitrarily deep nesting of triples as subject or object arguments:

*Definition 3.1.* An RDF* triple is a 3-tuple defined recursively as follows:

(1) Any RDF triple $t \in T$ is an RDF* triple; and
(2) Given RDF triples $t$ and $t'$, and RDF terms $s \in (I \times B)$, $p \in I$, and $o \in (I \times B \times L)$, then the tuples $(t, p, o)$, $(s, p, t)$, and $(t, p, t')$ are also RDF* triples.

In a nutshell, RDF* allows us to "say things" about RDF statements, which endows RDF with native *reification* capabilities. This is crucial when computing how-provenance for query results, because query provenance builds upon identifiers for triples in the graph.

RDF graphs can be queried using the SPARQL language. The fundamental building blocks of SPARQL queries are basic graph patterns (BGPs). They are sets of tuples of the form $(s, p, o) \in (V \cup I \cup B) \times (V \cup I) \times (V \cup I \cup B \cup L)$, where $V$ is a countably infinite set of variables—prefixed by the character '?'. Analogously to RDF*, SPARQL* extends BGPs by allowing nested patterns. Triple patterns and BGPs in SPARQL queries are combined with operators such as AND, UNION or SELECT. A (solution) *mapping* is a partial function $\mu : V \rightarrow (I \cup B \cup L)$ where the domain of $\mu$, denoted by $dom(\mu)$, is a finite set of variables. We write inScope($Q$) for the set of variables, called *in-scope*, that can occur in $Q$ answers, and variables that always occur are called *strongly bound* [4]. The details of SPARQL evaluation semantics are detailed in [2, 16]. In short, the evaluation of a SPARQL query $Q$ on an RDF graph $\Gamma$ is defined as a function $[\![Q]\!]_\Gamma$, which returns a multiset of mappings $\mu \in U$. Our goal is to annotate those mappings with their how-provenance.

## 3.2 How-provenance in SPARQL

### 3.2.1 Semirings.
A *commutative monoid* $\mathcal{M}$ is an algebraic structure $(M, +_{\mathcal{M}}, 0_{\mathcal{M}})$ such that $M \neq \emptyset$ is a set closed under a commutative and associate binary operation $+_{\mathcal{M}}$. The element $0_{\mathcal{M}}$ is the identity operand for $+_{\mathcal{M}}$. Given two commutative monoids $(K, +_{\mathcal{K}}, 0_{\mathcal{K}})$ and $(K, \times_{\mathcal{K}}, 1_{\mathcal{K}})$ such that $\times_{\mathcal{K}}$ is distributive over $+_{\mathcal{K}}$, and $0_{\mathcal{K}} \times_{\mathcal{K}} x = 0_{\mathcal{K}}$ (for every $x \in K$), we call the structure $\mathcal{K} = (K, +_{\mathcal{K}}, \times_{\mathcal{K}}, 0_{\mathcal{K}}, 1_{\mathcal{K}})$ a *commutative semiring*. An *spm-semiring* $(K, +_{\mathcal{K}}, \times_{\mathcal{K}}, -_{\mathcal{K}}, 0_{\mathcal{K}}, 1_{\mathcal{K}})$ extends a commutative semiring with a minus operation $-_{\mathcal{K}}$. This operator follows a set of axioms that allows us to model non-monotonic operations such as the relational difference. For more details about spm-semirings, we refer the reader to [11]. The algebraic expressions within an spm-semiring are used to annotate query solutions. To see how, we need to introduce the concepts of $\mathcal{K}$-relations and $\mathcal{K}$-graphs.

### 3.2.2 $\mathcal{K}$-relations and $\mathcal{K}$-graphs.
Given a set of mappings $U$ and an spm-semiring $\mathcal{K} = (K, +_{\mathcal{K}}, \times_{\mathcal{K}}, -_{\mathcal{K}}, 0_{\mathcal{K}}, 1_{\mathcal{K}})$, an annotation function $f : U \to K$ with finite support set $\text{supp}(f) = \{\mu \in U \mid f(\mu) \neq 0_{\mathcal{K}}\}$ is called a *$\mathcal{K}$-relation* over $U$. We call $f(\mu)$ the *$\mathcal{K}$-value* of $\mu \in U$. An annotation function $G : \Gamma \to K$ is called a *$\mathcal{K}$-graph*.

*Example 3.2.* Let $\mathcal{K} = (K, \oplus, \otimes, \ominus, 0, 1)$ be an spm-semiring with $K = \{u_1, u_2, u_3\}$, and let $G$ and $Q$ be the following RDF* graph and query:

$$G = \{ ((\textit{Alice, likes, pasta}), \textit{wasDerivedFrom}, u_1),$$
$$((\textit{Alice, likes, pasta}), \textit{wasDerivedFrom}, u_2),$$
$$((\textit{Alice, livesIn, Italy}), \textit{wasDerivedFrom}, u_3) \},$$

$$Q = (?x, \textit{likes, pasta}) \text{ AND } (?x, \textit{livesIn, Italy}).$$

It is easy to see that the predicate *wasDerivedFrom* defines a $\mathcal{K}$-graph $G : \Gamma \to K$, and that $\mu = \{?x \to \textit{Alice}\}$ is a solution mapping for $Q$. The set $\{\mu \to (u_1 \oplus u_2) \otimes u_3\}$ is a $\mathcal{K}$-relation that associates $Q$'s solution to a provenance polynomial. This how-provenance annotation tells us that *Alice* is a query solution for $Q$ as long as the triple identified by $u_3$ is present in conjunction with either the triples $u_1$ or $u_2$. We call those provenance identifiers the *sources*.

*Definition 3.3.* Given an spm-semiring $\mathcal{K}$, a SPARQL query $Q$ consisting of a combination of triple patterns with the operators AND, UNION, DIFF, FILTER, OPTIONAL, and SELECT, and a $\mathcal{K}$-graph $G$, we write $(\![P]\!)_G$ to denote the $\mathcal{K}$-relation defined recursively as follows:

$$(\![(s, p, o)]\!)_G(\mu) = G(\mu(s, p, o)),$$
$$(\![\text{SELECT } W \text{ WHERE } P]\!)_G(\mu) = \sum_{\mu'|_W = \mu} (\![P]\!)_G(\mu'),$$
$$(\![P \text{ FILTER } \varphi]\!)_G(\mu) = (\![P]\!)_G(\mu) \times_{\mathcal{K}} 1_{\mu \models \varphi},$$
$$(\![P_1 \text{ UNION } P_2]\!)_G(\mu) = (\![P_1]\!)_G(\mu) +_{\mathcal{K}} (\![P_2]\!)_G(\mu),$$
$$(\![P_1 \text{ AND } P_2]\!)_G(\mu) = \sum_{\mu = \mu_1 \cup \mu_2} ((\![P_1]\!)_G(\mu_1) \times_{\mathcal{K}} (\![P_2]\!)_G(\mu_2)),$$
$$(\![P_1 \text{ DIFF } P_2]\!)_G(\mu) = (\![P_1]\!)_G(\mu) -_{\mathcal{K}} (\sum_{\mu' \sim \mu} (\![P_2]\!)_G(\mu')),$$
$$(\![P_1 \text{ OPTIONAL } P_2]\!)_G(\mu) = (\![P_1 \text{ AND } P_2]\!)_G(\mu) +_{\mathcal{K}} (\![P_1 \text{ DIFF } P_2]\!)_G(\mu),$$

where $\sum$ denotes sums using the operation $+_{\mathcal{K}}$, and $\sim$ denotes mapping compatibility. Two mappings $\mu$ and $\mu'$ are compatible if $\mu(?x) = \mu'(?x)$ for every variable $?x \in \text{dom}(\mu) \cap \text{dom}(\mu')$.

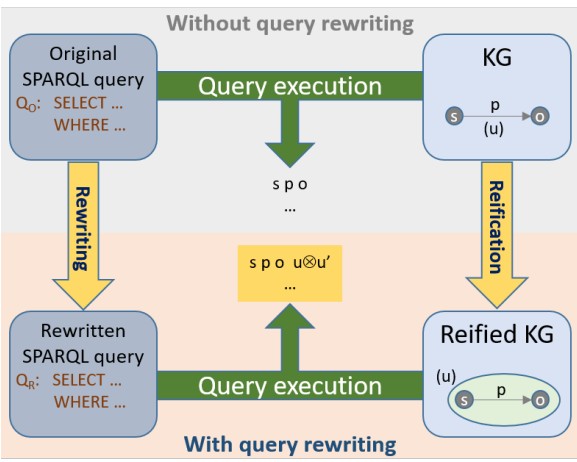

**Figure 1: The query rewriting process for NPCS.**

## 4 PROVENANCE COMPUTATION WITH NPCS

In the following, we explain our method to annotate SPARQL query solutions with how-provenance annotations as in Example 3.2. Given a $\mathcal{K}$-graph $G$ and a SPARQL query $Q$, NPCS rewrites $Q$ into $Q'$ so that the execution of $Q'$ returns a $\mathcal{K}$-relation that maps $Q$'s solutions to their how-provenance polynomials. Those polynomials lie in an spm-semiring $\mathcal{K} = (K, \oplus, \otimes, \ominus, 0, 1)$, where $K$ is the set of triple identifiers in $G$. We highlight that computing provenance assumes that the triples in the graph are reified, i.e., identified. RDF* is a natural way to do it, but as shown later, our approach supports any reification scheme for RDF data. The architecture of NPCS is depicted in Figure 1.

### 4.1 Our Query Rewriting in a Nutshell

Consider our $\mathcal{K}$-graph $G$ and query $Q$ from Example 3.2. For pedagogical reasons we rewrite our query $Q$ as $P_1$ AND $P_2$. We recall that our goal is to return the following result set:

$$\left[ \begin{array}{c|c} ?x & ?z \\ \hline \textit{Alice} & (u_1 \oplus u_2) \otimes u_3 \end{array} \right].$$

The column labeled $?z$ stores the how-provenance of each of the query solutions. To compute such an expression, our strategy must rewrite the query such that the rewritten query retrieves the identifiers of the triples that match each of the triple patterns. However, this is not enough. For instance, the triple pattern $P_1 = (?x, \textit{likes, pasta})$ has two matches, i.e., $u_1$ and $u_2$, that must be *grouped* into the expression $(u_1 \oplus u_2)$. This term tells us that the presence of at least one of those sources guarantees the inclusion of *Alice* in the result set. Finally, the groups extracted from each of the triple patterns must be combined with the $\otimes$ operator that explains the semantics of AND. We argue that to obtain the left-hand term of the product we can rewrite $P_1$ into the following sub-query:

$$P_1' = (\text{SELECT} \quad ?x \ (\text{ProvAggSum}(?z \oplus \otimes 1 \oplus \odot) \text{ AS } ?z \oplus \otimes 1)$$
$$\text{WHERE} \quad \text{Reify}(?x, \textit{likes, pasta}, ?z \oplus \otimes 1 \oplus \odot)$$
$$\text{GROUP BY} \quad ?x),$$

The Reify function rewrites a triple pattern so that it matches the reification scheme used in the $\mathcal{K}$-graph. In our running example, Reify is a shortcut for

$$((s, p, o), \textit{wasDerivedFrom}, ?z \oplus \otimes 1 \oplus \odot).$$

The intermediate variable $?z \oplus \otimes 1 \oplus \odot$ is introduced to capture the sources that match $P_1$, whereas $?z \oplus \otimes 1$ groups all the sources associated to a query solution, which explains the group clause on $?x$. The function ProvAggSum, later explained, combines the different sources into a summation with the operator $\oplus$.

The signs $\oplus$, $\otimes$, and $\odot$ in the intermediate variable names are strings used to produce new variable names that do not clash with the original query variables. The names of the variables encode the different steps of the construction of the annotations. For instance, the sign $\odot$ at the end of a variable name tells us that the variable's bindings are triple identifiers. If this is followed by a $\oplus$ sign, then those bindings will eventually be grouped into a summation by a subsequent step. Those results will be stored in a variable with the same prefix but without the suffix $\oplus \odot$. Furthermore, the $\otimes 1$ sign tells us that our results correspond to the first operand of a join operation, namely AND in SPARQL. If we apply the same logic to $P_2$ our rewriting for $Q$ takes the following form:

$$Q' = (\;\text{SELECT} \quad ?x\;(\text{ProvAggSum}(?z\oplus)\;\text{AS}\;?z)$$
$$\text{WHERE} \quad ((P_1\;\text{AND}\;P_2)$$
$$\text{BIND}\;(\text{ProvProd}(?z\oplus\otimes 1, ?z\oplus\otimes 2)\text{AS}\;?z\oplus))$$
$$\text{GROUP BY} \quad ?x\;),$$

The operation ProvProd combines the expressions derived from the product's operands. We resort again to ProvAggSum to sum up all the ways to produce a solution mapping from a join operation. The operators ProvAggSum and ProvProd are defined in terms of the standard SPARQL string operators `concat` and `aggregate_concat` as follows.

*Definition 4.1.* Let $?x, ?x_1, \dots, ?x_n$ be variables. Then, we define the following SPARQL operators:

$\text{ProvAggSum}(?x) = \text{concat}(``(\oplus", \text{aggregate\_concat}(?x), ``)"),$
$\text{ProvSum}(?x_1, \dots, ?x_n) = \text{concat}(``(\oplus", ?x_1, \dots, ?x_n, ``)"),$
$\text{ProvProd}(?x_1, \dots, ?x_n) = \text{concat}(``(\otimes", ?x_1, \dots, ?x_n, ``)"),$
$\text{ProvDiff}(?x_1, ?x_2) = \text{concat}(``(\ominus", ?x_1, ?x_2, ``)").$

## 4.2 Base Rewriting Rules

Having provided the intuition behind our query rewriting in Section 4.1, we now introduce our rewriting rules for arbitrary SPARQL queries.

*Definition 4.2 (Base SPARQL* query rewriting).* Let $Q$ be a SPARQL query, $?z$ a variable, and Reify a reification scheme. Then, the rewritten query for $Q$ and variable $?z$ over scheme Reify, denoted $\beta(Q, ?z)$, is defined recursively as follows:

(1) If $Q$ is an empty basic graph pattern, then $\beta(Q, z)$ is the query $\{\}$ BIND $(1\;\text{AS}\;?z)$.

(2) If $Q$ is a triple pattern $(s, p, o)$, then $\beta(Q, z)$ is the query
$(\;\text{SELECT} \quad \text{inScope}(Q)\;(\text{ProvAggSum}(?z\oplus\odot)\;\text{AS}\;?z)$
$\text{WHERE} \quad \text{Reify}(p(s, o), ?z\oplus\odot)$
$\text{GROUP BY} \quad \text{inScope}(Q)\;).$

(3) If $Q$ is $(P_1\;\text{AND}\;P_2)$, then $\beta(Q, ?z)$ is the query
$(\;\text{SELECT} \quad \text{inScope}(Q)\;(\text{ProvAggSum}(?z\oplus)\;\text{AS}\;?z)$
$\text{WHERE} \quad (\beta(P_1, ?z\oplus\otimes 1)\;\text{AND}\;\beta(P_2, ?z\oplus\otimes 2))$
$\qquad\qquad \text{BIND}\;(\text{ProvProd}(?z\oplus\otimes 1, ?z\oplus\otimes 2)\;\text{AS}\;?z\oplus)$
$\text{GROUP BY} \quad \text{inScope}(Q)\;).$

(4) If $Q$ is $(P_1\;\text{UNION}\;P_2)$, then $\beta(Q, ?z)$ is the query
$(\;\text{SELECT} \quad \text{inScope}(Q)\;(\text{ProvAggSum}(?z\oplus)\;\text{AS}\;?z)$
$\text{WHERE} \quad (\beta(P_1, ?z)\;\text{UNION}\;\beta(P_2, ?z))$
$\text{GROUP BY} \quad \text{inScope}(Q)\;).$

(5) If $Q$ is $(P_1\;\text{DIFF}\;P_2)$, then let $\nu$ a variable substitution that substitutes with fresh variables the variables in $\text{dom}(P_1) \cap \text{dom}(P_2)$ that are not strongly bound in $P_1$. Then, $\alpha(Q, ?z)$ is the query computed as follows.
$(\;\text{SELECT} \quad \text{inScope}(Q)$
$\qquad\qquad (\text{ProvDiff}(z\ominus 1, \text{ProvAggSum}(z\ominus 2\oplus))\;\text{AS}\;?z)$
$\text{WHERE} \quad (\beta(P_1, ?z\ominus 1)\;\text{OPTIONAL}_{C_\nu}\;\beta(\nu(P_2), ?z\ominus 2\oplus))$
$\text{GROUP BY} \quad \text{inScope}(Q) \cup \{z\ominus 1\}\;).$

(6) If $Q$ is $(\text{SELECT}\;W\;\text{WHERE}\;P_1)$, then $\beta(Q, ?z)$ is the query
$(\;\text{SELECT} \quad W\;(\text{ProvAggSum}(?z\oplus)\;\text{AS}\;?z)$
$\text{WHERE} \quad \beta(P_1, ?z\oplus)$
$\text{GROUP BY} \quad W\;).$

(7) If $Q$ is $(P\;\text{FILTER}\;\varphi)$, then $\beta(Q, ?z)$ is the query
$(\beta(P, ?z)\;\text{FILTER}\;\varphi).$

(8) If $Q$ is $(P\;\text{BIND}\;(E\;\text{AS}\;?x))$, then $\beta(Q, ?z)$ is the query
$(\beta(P_1, ?z)\;\text{BIND}\;(E\;\text{AS}\;?x)).$

We omit the rewriting rule for the OPTIONAL operator because this operator can be written in terms of AND, UNION and DIFF, to be precise, $P_1\;\text{OPTIONAL}\;P_2 \equiv (P_1\;\text{DIFF}\;P_2)\;\text{UNION}\;(P_1\;\text{AND}\;P_2)$.

Since the DIFF operation requires tracking the provenance of both operands, DIFF translates to an OPTIONAL operation, more precisely, to an $\text{OPTIONAL}_{C_\nu}$ operation, which extends the OPTIONAL operation by renaming variables in the optional pattern with fresh ones. This renaming discards undesired bindings produced in the subtrahend while tracking the provenance. For example, consider the patterns $P_1(?x, ?y)$ and $P_2(?x, ?y, ?z)$, whose in-scope variables are indicated in the parenthesis, and let $\mu_1 = \{?x \mapsto a\}$ and $\mu_2 = \{?x \mapsto a, ?y \mapsto b, ?z \mapsto\}$ be the respective results of them. If instead of $\text{OPTIONAL}_{C_\nu}$ rule (5) uses OPTIONAL, the query will return the provenance for the mapping $\{?x \mapsto a, ?y \mapsto b\}$ instead of the mapping $\mu_1$. If $?x$ is strongly bound for $P_1$ (i.e., always bound in its answers), then the operation $(P_1\;\text{OPTIONAL}_{C_\nu}\;P_2$ will be:

$$((P(?x, ?y)\;\text{OPTIONAL}\;P(?x, \nu(?y), ?z))$$
$$\text{FILTER}\;(\neg\,\text{bound}(?y) \vee \neg\,\text{bound}(\nu(?y)) \vee ?y = \nu(?y))).$$

To define correctness of the query rewriting described in Definition 4.2 we need the notions of *soundness* and *completeness*. A query rewriting is sound when the rewritten query returns right polynomial expressions, and complete if it returns polynomial expressions for all mappings with non-zero polynomials.

*Definition 4.3 (Soundness and Completeness).* Let Reify be a function that implements a reification scheme, and $\gamma$ be a function that receives a SPARQL query $Q$ and a variable $?z \notin \text{inScope}(Q)$, and returns a SPARQL query $\gamma(Q, z)$ with $\text{inScope}(\gamma(Q, ?z)) = \text{inScope}(Q) \cup \{?z\}$. Let $G$ be a $\mathcal{K}$-annotated graph, and $\text{Reify}(G)$ the RDF-star graph resulting from applying the Reify function to each triple in $G$ in order to encode $G$'s triple annotations.

(1) $\gamma$ is called *sound* for Reify if, for every answer of the rewritten query $\mu \cup \{?z \mapsto e\} \in [\![\gamma(Q, ?z)]\!]_{\text{Reify}(G)}$, $e$ is an expression for the polynomial $(\![Q]\!)_G(\mu)$.

(2) $\gamma$ is called *complete* for Reify if, for every mapping $\mu$ such that $(\![Q]\!)_G(\mu)$ is a non-zero polynomial, there exists an expression $e$ such that $\mu \cup \{?z \mapsto e\} \in [\![\gamma(Q, ?z)]\!]_{\text{Reify}(G)}$.

THEOREM 4.4. *Let* Reify *be a reification scheme, and* $\beta$ *be the function described in Definition 4.2. Then, function* $\beta$ *is sound and complete for the reification scheme* Reify.

PROOF. It can be shown by induction on the query structure. □

We highlight that for queries of the form $P_1$ DIFF $P_2$, NPCS also returns why-not provenance explanations of the form $k_1 \ominus k_2$ ($k_1$ and $k_2$ are polynomials) for the bindings that match both $P_1$ and $P_2$. The polynomial $k_2$ tells us which sources must be removed from the graph so that the corresponding binding becomes a query solution.

## 4.3 Query Rewriting Optimization

If we look at our example rewritten query $Q'$ described in Section 4.1, we can notice that this query includes a GROUP BY clause, and two subqueries, namely $P_1'$ and $P_2'$, each of which also includes a GROUP BY clause. In Definition 4.6 we describe an alternative query rewriting that produces equivalent polynomial expressions, but reduces the number of aggregate operations.

*Definition 4.5.* A *sum-query* $Q$ is a query such that the query rewriting $\beta$ described in Definition 4.2 returns a query of the form:

$$\beta(Q, ?z) = (\text{SELECT} \quad \text{inScope}(Q) \; (\text{ProvAggSum}(?z\oplus) \text{ AS } ?z)$$
$$\text{WHERE} \quad T$$
$$\text{GROUP BY} \quad \text{inScope}(Q) \,).$$

We call query $T$ the *pattern* of $\beta(Q, ?z)$.

Note that, according to Definition 4.2, sum-queries are all queries that match the rules 2, 3, 4, and 6.

*Definition 4.6.* Let $Q$ be a SPARQL query, $?z$ be a variable, and Reify a reification scheme. Then, the rewritten query for $Q$ and variable $?z$ over scheme Reify, denoted $\beta(Q, ?z)$, is defined recursively as is specified in Definition 4.2, but the following rules are applied when possible:

(1) If $Q$ is $(P_1 \text{ AND } \cdots \text{ AND } P_n)$, and $P_1, \ldots, P_n$ are sum-queries, such that for $1 \leq i \leq n$, the pattern of $\beta(P_i, ?z\oplus\otimes i)$ is $T_i$, then $\beta(Q, ?z)$ is the query

$$(\text{SELECT} \quad \text{inScope}(Q) \; (\text{ProvAggSum}(?z\oplus) \text{ AS } ?z)$$
$$\text{WHERE} \quad (( T_1 \text{ AND } \cdots \text{ AND } T_n)$$
$$\text{BIND} \; (\text{ProvProd}(?z\oplus\otimes 1\oplus, \ldots ?z\oplus\otimes n\oplus)$$
$$\text{AS } z\oplus)$$
$$\text{GROUP BY} \quad \text{inScope}(Q) \,).$$

(2) If $Q$ is $(P_1 \text{ UNION } \cdots \text{ UNION } P_n)$, where $P_1, \ldots, P_n$ are sum-queries, and for $1 \leq i \leq n$, the pattern of $\beta(P_i, ?z\oplus)$ is $T_i$, then $\beta(Q, ?z)$ is the query

$$(\text{SELECT} \quad \text{inScope}(Q) \; (\text{ProvAggSum}(?z\oplus) \text{ AS } ?z)$$
$$\text{WHERE} \quad (T_i \text{ UNION } \cdots \text{ UNION } T_n)$$
$$\text{GROUP BY} \quad \text{inScope}(Q) \,).$$

(3) If $Q$ is $(P_1 \text{ DIFF } P_n)$, and $\nu$ is a variable substitution that substitutes with fresh variables the variables in $\text{dom}(P_1) \cap \text{dom}(P_2)$ that are not strongly bound in $P_1$, and $P_2$ is a

sum-query where the pattern of $\beta(\nu(Q_2), ?z\ominus 2)$ is $T_2$, then $\beta(Q, ?z)$ is the query

$$(\text{SELECT} \quad \text{inScope}(Q)$$
$$\quad\quad\quad (\text{ProvDiff}(z\ominus 1, \text{ProvAggSum}(z\ominus 2\oplus)) \text{ AS } ?z)$$
$$\text{WHERE} \quad (\beta(P_1, ?z\ominus 1) \text{ OPTIONAL}_{C_\nu} T_2)$$
$$\text{GROUP BY} \quad \text{inScope}(Q) \cup \{z\ominus 1\} \,).$$

(4) If $Q$ is $(\text{SELECT } W \text{ WHERE } P)$, where $P$ is a sum-query such that the pattern of $\beta(P, ?z\oplus)$ is $T$, then $\beta(Q, ?z)$ is the query

$$(\text{SELECT} \quad W \; (\text{ProvAggSum}(?z\oplus) \text{ AS } ?z)$$
$$\text{WHERE} \quad T$$
$$\text{GROUP BY} \quad W \,).$$

Note: In rules 1 and 2 of Definition 4.6, we omitted the parenthesis for sequences of operations AND and UNION, because these operators are associative. Intuitively, the associativeness of these operators allows considering the binary operation as a single variadic operation with a single GROUP BY clause.

*Example 4.7.* Consider the query $Q$ from Example 3.2. Then, according to the query rewriting described in Definition 4.6 the rewritten query of $Q$ is:

$$\beta(Q, ?z) = (\text{SELECT} \quad ?x \; (\text{ProvAggSum}(?z\otimes\otimes) \text{ AS } ?z)$$
$$\text{WHERE} \quad (( \text{Reify}(?x, \text{likes}, \text{pasta}, ?z\oplus\otimes 1\oplus) \text{ AND}$$
$$\quad\quad\quad \text{Reify}(?x, \text{likes}, \text{pasta}, ?z\oplus\otimes 2\oplus))$$
$$\quad\quad\quad \text{BIND} \; (\text{ProvProd}(?z\otimes\otimes 1\oplus, ?z\otimes\otimes 2\oplus)$$
$$\quad\quad\quad\quad \text{AS } ?z\oplus\otimes))$$
$$\text{GROUP BY} \quad ?x \,).$$

This query has only one GROUP BY clause whereas the query $Q'$ at the end of Section 4.1 (generated using the rewriting of Definition 4.2) has three.

THEOREM 4.8. *Let* Reify *be a reification scheme, and* $\beta$ *be the function described in Definition 4.6. Then, function* $\beta$ *is sound and complete for the reification scheme* Reify.

PROOF. It can be shown by induction on the query structure, that the expressions resulting from the rules in Definition 4.6 produce the same results that the rewriting in Definition 4.2. □

## 5 EVALUATION

We conducted an extensive evaluation of NPCS's viability for computing how-provenance by assessing the runtime overhead incurred by the rewritten queries. This is measured by comparing the runtime between the original queries without provenance annotations and the queries obtained with our approach.

## 5.1 Experimental Setup

*5.1.1 Environment.* NPCS was implemented in Java, using the Java Development Kit (JDK) version 11. All the experiments were conducted on a computer with an AMD EPYC 7281 16-core processor, 256GB of RAM, and an 8 TB HDD disk running Ubuntu 18.04.6 LTS. We evaluated NPCS on two widely used RDF/SPARQL engines, namely GraphDB[1] (version 10.2.0) and Stardog[2] (version 9.1.0). Throughout all experiments, we set a timeout of 350 seconds to ensure consistent results and reported the average response time of the queries over five executions in a cold setting, i.e., after having cleared the disk cache.

---

[1] https://graphdb.ontotext.com/
[2] https://www.stardog.com/

*5.1.2 Competitor.* We compare NPCS with SPARQLprov [15], a state-of-the-art solution for how-provenance in SPARQL, which is also based on query rewriting. We used the implementation provided by the authors of [15], except from the RDF* reification scheme, which was not supported by the original SPARQLprov implementation, and we implemented it ourselves. SPARQLprov and NPCS compute the same provenance polynomials since they both rely on spm-semirings.

*5.1.3 Synthetic Workload.* We employed the Watdiv performance benchmark specifically designed for RDF/SPARQL engines. Watdiv provides a data generator that can produce synthetic datasets of varying sizes. Additionally, WatDiv includes 20 SELECT query templates, each comprising 10 instantiated queries. The query templates are categorized into four types: linear queries (L), star queries (S), snowflake-shaped queries (F), and complex queries (C). They are all monotonic queries. We therefore introduced five additional non-monotonic query templates (O) as proposed by [15]. These non-monotonic queries were created by enclosing one of the triple patterns in the linear queries with an OPTIONAL clause.

We evaluate NPCS on the 10M-triple and 100M-triple Watdiv synthetic datasets, that we reified using the RDF* and named graphs reification schemes. We excluded the standard reification from our evaluation as it reports the worst performance according to [15]. Moreover, we created a 200M-triple dataset by duplicating every triple of the 100M-triple dataset and assigning a second provenance identifier to the duplicates. This dataset aims at simulating a challenging scenario where triples has been extracted from more than one source.

*5.1.4 Real Workload.* We tested NPCS and SPARQLprov on the WDBench benchmark [1] in order to evaluate our methodology on real-world data. The benchmark uses data from 15.2 billion triples encoded using the Wikidata reification scheme in the Wikidata dump from 2023. The benchmark provides more than 800 queries consisting of simple BGPs, some of them with OPTIONAL clauses. We took a sample of 150 queries consisting of 50 single-triple-pattern queries, 50 non-monotonic queries (with OPTIONAL), and 50 monotonic queries with more than on triple pattern. The queries were randomly chosen.

## 5.2 Results

*5.2.1 Synthetic Workload.* Figures 2 and 3 compare the execution times of the original query with those of the rewritten queries produced by NPCS and our competitor SPARQLprov on RDF* data when using GraphDB and Stardog respectively. The runtimes were measured on the 10M and 100M Watdiv synthetic datasets. We first notice that in all cases, rewriting the query to compute how-provenance incurs a performance overhead. Not surprisingly, the overhead increases with data size, but its behavior also depends on the query engine. For instance, NPCS's overhead ranges from 20% to 30% in GraphDB, and from 25% to 50% in Stardog. Figures 2 and 3 also reveal that, on average, GraphDB is more sensitive to data scaling than Stardog, even though runtime across query templates exhibits higher variability in Stardog. Regardless of the data size and the query engine, templates C3 and F5 are by far the most challenging, which makes our competitor SPARQLprov timeout on

both GraphDB and Stardog. The complexity of C3 is explained by its large number of intermediate results, whereas for F5 it is caused by the large number of solutions.

When we compare the query rewriting strategies, we notice the NPCS consistently outperforms SPARQLprov in 98 out of our 100 studied cases. Also, NPCS is on average 25 times faster than SPARQLprov. One can explain this performance difference by the fact that NPCS is a fully native SPARQL solution, whereas SPARQLprov relies on a post-hoc decoding phase to compute the provenance polynomials. Like NPCS, SPARQLprov rewrites the query to extract provenance information. Unlike our approach, SPARQLprov encodes the structure of the how-provenance annotations in additional columns in the result set. Those additional columns can be numerous and encode the structure of the provenance polynomials. Decoding that information requires to run additional group and aggregation operations. Hence, the runtime of this decoding phase is proportional to the number of query solutions times the maximal depth of the operator trees of the provenance annotations. That explains why SPARQLprov times out for query template F5, which is by far the template with the highest number of query solutions (173.6K solutions on average). NPCS, in contrast, carries out the grouping operations during query evaluation, which not only leverages the engine optimizations for grouping but also makes it easier to deploy in real-world settings.

Despite NPCS's clear runtime advantage, SPARQLprov can exhibit comparable or better performance on very selective queries. This is demonstrated by the runtimes for queries O1, O2, and O5. In cases such as query template O5, and query template O2 on GraphDB, NPCS's strategy of evaluating grouping operations in the source engine, does not pay off. This is so because the queries and their constituent triple patterns are very selective.

We now shift our attention to the runtime results on the 200M dataset. The results for our studied rewriting methods are depicted in Figure 4 on GraphDB. We observe similar trends as in the 100M scenario, except that SPARQLprov also times out on query templates C2 and F4. We omit the results for Stardog as they exhibit similar behavior as in the 100M dataset.

Finally we evaluate NPCS on a different reification scheme, namely the popular named graphs strategy. The results are depicted in Figure 5 for the 10M and 100M datasets on GraphDB. We observe the same trends as for the RDF* reification, that is, NPCS outperforms SPARQLprov consistently in 48 out of 50 studied cases. This shows that our approach is insensitive to the data reification scheme, which makes it applicable to any standard RDF/SPARQL engine. Similar results are observed for Stardog.

*5.2.2 Real Workload.* We now evaluate NPCS on real-world data, namely on the WDBench based on Wikidata. The results for GraphDB and Stardog are shown in Figures 6 and 7 respectively. Each dot in the plot represents the execution of a query, either the original query or a rewritten query by NPCS or by SPARQLprov. Queries are plotted on the x-axis by increasing number of solutions, whereas the y-axis represents the execution time. We verify the same trend for both engines, namely that SPARQLprov's query rewriting induces a much larger overhead than NPCS. While the overhead increases with the number of query results for both

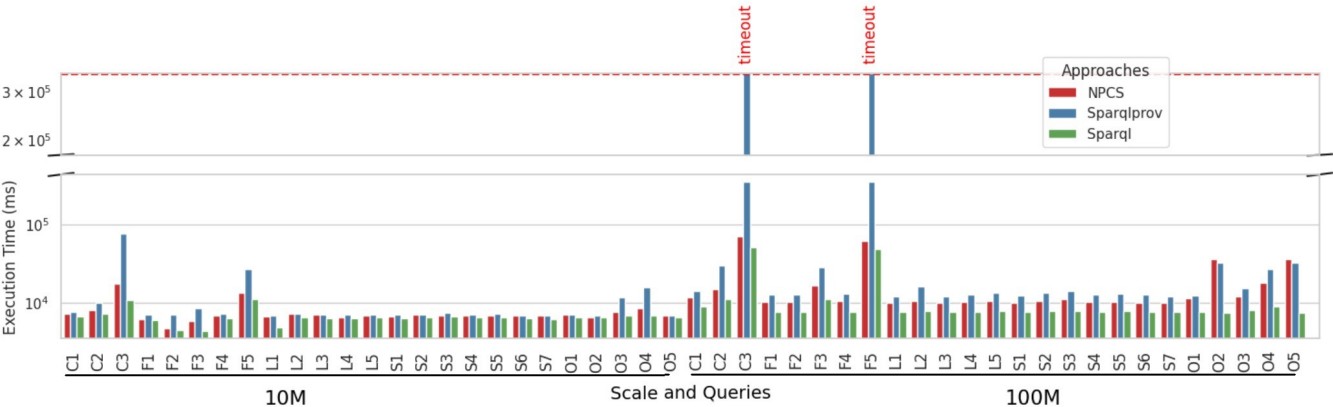

Figure 2: Query execution times on the Watdiv 10M and 100M datasets reified with RDF* on GraphDB

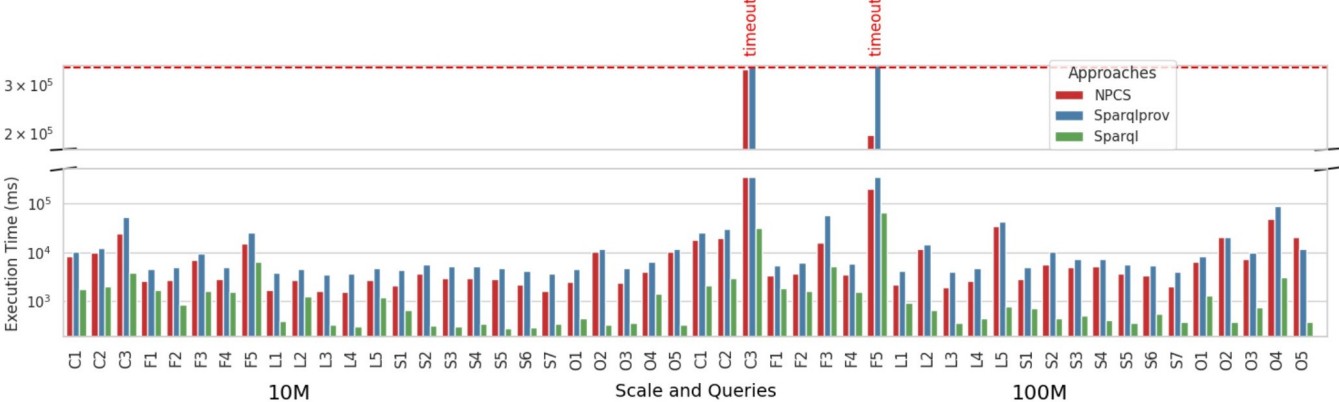

Figure 3: Query execution times on the Watdiv 10M and 100M datasets reified with RDF* on Stardog

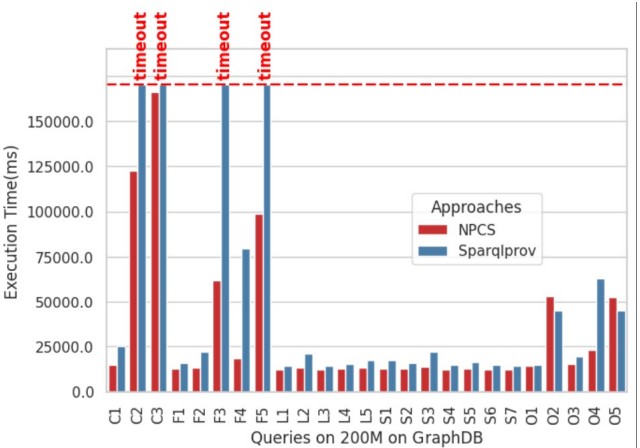

Figure 4: Query execution times on the Watdiv 200M dataset reified with RDF* on GraphDB

methods, it is more pronounced for SPARQLprov. This makes SPARQLprov time out when the number of results is above 700K. We also observe that for queries with a few thousand results executed on Stardog, NPCS's overhead can be minimal.

## 6 CONCLUSIONS

In this paper we have proposed NPCS, a novel query rewriting method to compute how-provenance annotations for SPARQL query results. To the best of our knowledge, NPCS is the first 100% SPARQL-based solution for how-provenance. Our approach can be easily applied on standard and already deployed RDF/SPARQL engines, without the need for customized extensions or post-processing steps. Our experimental evaluation on synthetic and real data shows that NPCS's native SPARQL rewriting outperforms the state of the art in how-provenance for SPARQL queries. The performance gains provided by our method allows us to compute provenance annotations for millions of query results on knowledge graphs with billions of triples. This makes our approach attractive for ETL processes that manage large volumes of data—a common scenario for multi-source KG construction.

As future work we intend to work on lazy approaches for how-provenance computation, that is, approaches where provenance is

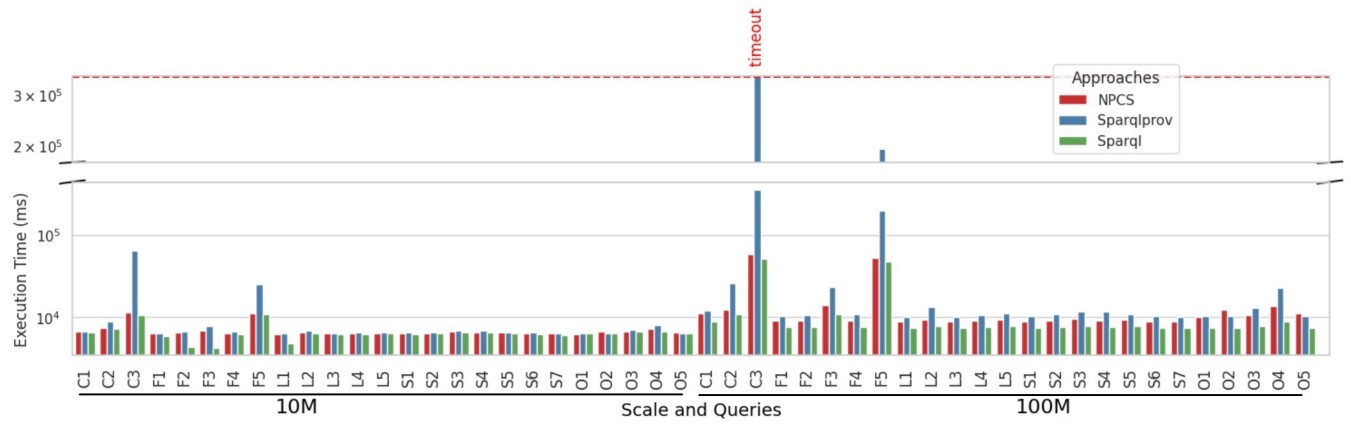

**Figure 5: Query execution times on the Watdiv 10M and 100M datasets reified as named graphs on GraphDB.**

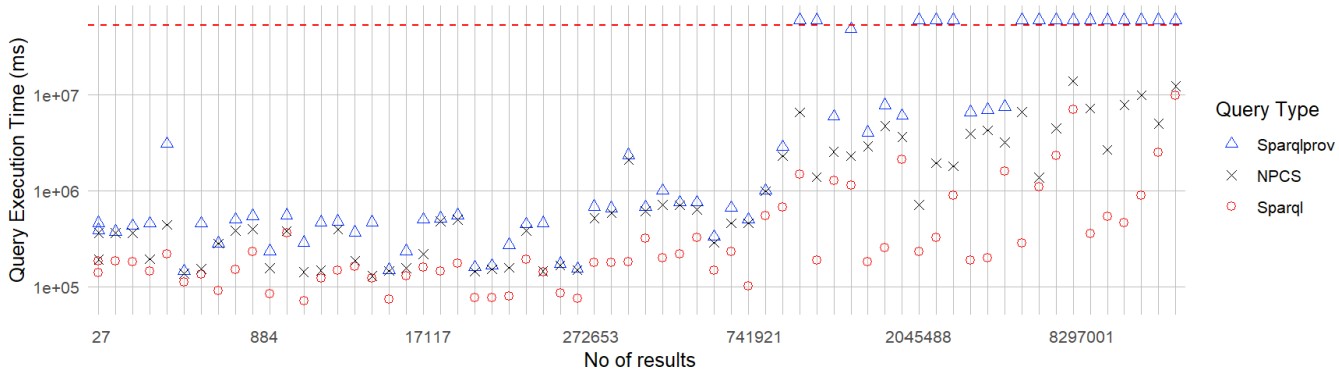

**Figure 6: Number of results vs. query execution time for the WDBench queries run on Wikidata (stored in GraphDB using the Wikidata reification scheme)**

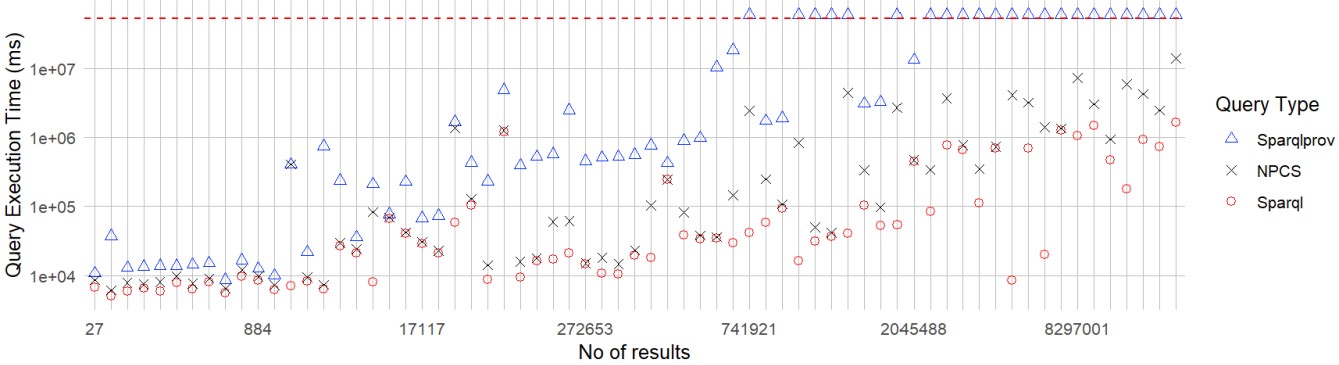

**Figure 7: Number of results vs. query execution time for the WDBench queries run on Wikidata (stored in Stardog using the Wikidata reification scheme)**

computed for a user-specified set of solutions. This avoids the execution of expensive queries for results that are not of interest of the user. We have also envisioned to tackle the problem of computing how-provenance annotations for non-reified data.

## SUPPLEMENTARY MATERIAL STATEMENT

The source code of NPCS, the scripts to recreate the full experimental setup, required libraries, queries, and results can be found on GitHub.[3]

---

[3]URL: https://github.com/factcheckerr/NPCS

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
