# OpenReview forum: "NPCS: Native Provenance Computation for SPARQL"
_ACM.org/TheWebConf/2024/Conference — TheWebConf24 Oral_

### Official Review · Reviewer_9XuD · 2023-11-23

**Novelty:** 5
**Technical Quality:** 6

**Review:**

In the context of RDF Knowledge Graphs, the paper aims to add “how-provenance” on SPARQL query results. How-provenance is important for downstream tasks such as view-maintenance, reasoning, trust-assessment, source selection. The “how-provenance” is computed relying on semi-rings. Semi-rings have been introduced in 2007 in the database field, and adapted recently to SPARQL to handle non-monotonic operators such as “OPTIONAL”.
The main contribution of NPCS is to provide “how-provenance” with pure rewriting of SPARQL queries using reification. The main advantage of NCPS compared to previous approaches is to run on current  engines as it is with no post-process of results as in [15].

Strong points:
* Having how-provenance on the current SPARQL engine is an important improvement for SPARQL, unlocking many downstream tasks.
* The paper is well-written and understandable.
* The positioning vs related works is very clear.
* The proposal explains clearly the rewriting rules.
* The methodology of experiments is ok.

Weak points:
* If the impact of the proposal is clear, the scientific contribution compared to [15] remains unclear to me. Compared to [15], the only thing we know is “Contrary to NPCS, SPARQLprov is not a 100% SPARQL solution, because it relies on a subsequent decoding phase to compute the final provenance annotations from the results of the rewritten query”. In fact we don’t know what has been done to remove the “subsequent decoding phase” of [15]. Was it challenging, or just a pb of implementation? To be clear, the impact can be high, but the scientific contribution can be minor, more explanations are needed on this point.

* In experiments, in [15] SPARQLPRov was compared to tripleprov. Why tripleprov is no longer present in these experiments. As performances always matters, and tripleprov can be fast thanks to its dedicated engine, we need to know if NPCS dominates TripleProv or not, even if TripleProv and NCPS does not belongs to the same category (rewriting vs specialised engine)

* It is written “The benchmark uses data from 15.2 billion triples encoded using the Wikidata reification scheme in the Wikidata dump from 2023”. Ok, but WDbench has just 1,257,169,959 triples, maybe it should be stated clearly the size of the dataset you really used. One missing point is the size of datasets once reified compared to the size of the dataset with no reification.

* It is not clear what is the green “Sparql” bar in experimental results. Is it the execution of original SPARQL queries with no how-provenance rewriting, but on which data: reified or not reified? I think it is important to understand the overhead of the “how-provenance” in terms of data size and execution time.

* Why is there no “SPARQL” green bar on Figure 4 ??

* I don’t understand why the plotting has changed for figure 6 and 7. I think it is more difficult to read than the previous figures. Is it possible to replot figure 6/7  as figure 5?

* On github, I cannot see the rewritten queries. Is there a way to see them without running your code ? (i tried with docker image but i got: % docker run --name npcs_container npcs:v1 SPARQL_Star queries\watdivqueries\Basic\C1\00.sparql
/usr/local/bin/mvn-entrypoint.sh: 50: exec: SPARQL_Star: not found)

**Questions:**

* In fact we don’t know what has been done to remove the “subsequent decoding phase” of [15]. Was it challenging, or just a pb of implementation? To be clear, the impact can be high, but the scientific contribution can be minor, more explanations are needed on this point.

* Why tripleprov is no longer present in these experiments ?

* On github, I cannot see the rewritten queries. Is there a way to see them without running your code (see the github issue)?

**Reviewer Confidence:**

4: The reviewer is certain that the evaluation is correct and very familiar with the relevant literature

**Scope:**

4: The work is relevant to the Web and to the track, and is of broad interest to the community

---

### Official Review · Reviewer_kp7h · 2023-11-24

**Novelty:** 6
**Technical Quality:** 5

**Review:**

The paper proposes NPCS, a novel query rewriting method for how-provenance annotations for SPARQL query results. NPCS can be directly applied to already deployed SPARQL engines using different reification schemes, including RDF*. As NPCS builds upon spm-provenance semirings, it supports monotonic and non-monotonic SPARQL queries. The experimental evaluation on two popular SPARQL
engines (GraphDB and Stardog) show that NPCS query rewriting improves runtime significantly over existing query rewriting solutions, scaling to RDF graphs with billions of triples.

## Strength

- Important problem.
- The paper is well-written and easy to follow.
- Elegant solution.
- Reproducibility.

## Weakness

- Some rewritings are unclear (see questions)
- Unclear why it is challenging.
- Used engines in experimental study.


Minors: missing parenthesis in : (𝑃1 optional𝐶𝑣 𝑃2

**Questions:**

## Questions:

Q1: Compared to [15], NPCS does not require post-processing. Why is it difficult to eliminate the post-processing? Why is it challenging?

Q2: An annotation function 𝐺 : Γ → 𝐾 is called a K-graph, what Γ refer to?

Q3: What does inScope refer to?

Q4: Dealing with OPTIONAL rewriting is an important contribution to the paper. However, I am confused with the OPTIONAL rewriting.
According to the paper, the rewriting of the optional is omitted because optional can be written in terms of AND, UNION, and DIFF (I agree with this). After that, it is written that DIFF is translated into OPTIONAL (OPTIONALCv). Please clarify this.

Q5: Another problem with DIFF is that no operator explicitly named DIFF  in the SPARQL query language. However, it is possible to simulate DIFF using MINUS or NOT EXISTS, which could produce different results. Could you elaborate on this?

Q6: Why does the experimental study use GraphBD and Stardog? Why are Fuseki and Virtuous not used in the evaluation? Are there any special requirements for the implementation of NPCS available in GraphBD and Stardog and not in Fuseki or Virtuouos?

Q7: Does the approach work for Aggregate queries?

**Reviewer Confidence:**

4: The reviewer is certain that the evaluation is correct and very familiar with the relevant literature

**Scope:**

4: The work is relevant to the Web and to the track, and is of broad interest to the community

---

### Official Review · Reviewer_A4it · 2023-11-29

**Novelty:** 3
**Technical Quality:** 4

**Review:**

In this paper proposes a query rewriting method to compute how-provenance annotations for SPARQL query results.  It is 100% SPARQL-based solution for how-provenance and can be easily applied on standard and already deployed RDF/SPARQL
engines.  Evaluation is performed both in real and synthetic datasets. The evaluation results are promising comparing state of the art.

The paper reads well. The topic is relevant to the conference. It deals an interesting and important topic. My only question (except for post processing)  how it is different from [15]? I would like to know the key differences from the core approach/implementation point of view.

**Questions:**

Q1. Except for the requirement of postprocessing? How the proposed method is different from [15] ?

**Ethics Review Description:**

Nothing

**Reviewer Confidence:**

2: The reviewer is willing to defend the evaluation, but it is likely that the reviewer did not understand parts of the paper

**Scope:**

4: The work is relevant to the Web and to the track, and is of broad interest to the community

---

### Official Review · Reviewer_BUtq · 2023-11-30

**Novelty:** 5
**Technical Quality:** 5

**Review:**

The paper “NPCS: Native Provenance Computation for SPARQL” provides a a mechanism and an implementation to retrieve How-provenance for SPARQL queries. Th mechanism is  only relies on SPARQL and does not depend on external calculations. The implementation is based on RDF-star and SPARQL-star which are used to encode and query reified triples. The authors compare their implementation to a modified version of the state-of-the-art implementation SPARQLProv and show that their engine performs significantly faster than SPARQLProv.

While the topic of the paper is certainly interesting, especially the practical aspect on how RDF-star could help computing provenance (also considering that the original SPARQLProv shows that the reification mechanism actually matters), I see several issues that should be improved before publication.

I do like that the paper comes with a formalization, but this formalization needs to be done more thoroughly. Some definitions seem to be wrong, others are underspecified. I cannot fully conclude from formalization and examples how the provenance calculation is done (even though I still get an idea and the idea looks promising). I would furthermore be interested in the concrete representation of K-representations in RDF-star since that is a crucial part of the implementation. I was also curious about the concrete SPARQL-star queries which were used but I could not even find examples in the git repository. I think that these kinds of details would make the paper more interesting.

Another weak point of the paper is that the evaluation relies on a rewriting of SPARQLProv which is not properly explained. I need to believe the authors that we have a fair comparison, but this would be easier if they provided more details on the changes they made on the original implementation.

To summarize: The topic of the paper is interesting and also relevant for the conference, but the paper needs a significant amount of work to flesh out the details.


Further comments and typos:

- the paper has AAAI in its header which suggests that it was rejected there in the first round. While this is not per-se a problem, I still think that as a matter of respect to the reviewers,  the authors could have at least changed the header to make this fact less obvious.

- rdf* and sparql* where renamed to rdf-star and sparql-star

- there is no reference for sparql-star, the semantics is currently under development, therefore it is important to note which version the paper uses

-  line 89 order -> other

- line 207-208: why are subject, predicate and object suddenly tuples?

- G on line 249 ff: The G seems to be a mapping and a graph at the same time, that needs explanation

- line 253: what is \Gamma? Is that a graph? Annotation functions are defined for mappings . So how does it work here?

- line 275: what is P? a query? only Q is introduced.

- line 279: what does the sum for the select pattern mean? how can the set W be equal to the mapping \mu?

- line 354: The variable naming is very confusing. I recommend not using special signs in variable names and keep these simple. It is still possible to explain the variable use as you do it in the text without adding confusion about the meaning of the signs.

- line 475: I would have liked to see the proof, if it was omitted for space issues, it could have gone to some appendix.

**Questions:**

- Would it be possible to get a link to a concrete query in the repository?
- I know that this is a broad question, but I would be interested in some clarification regarding the formalization.

**Ethics Review Description:**

nothing

**Reviewer Confidence:**

3: The reviewer is confident but not certain that the evaluation is correct

**Scope:**

4: The work is relevant to the Web and to the track, and is of broad interest to the community

---

### Official Review · Reviewer_kkQe · 2023-12-01

**Novelty:** 6
**Technical Quality:** 7

**Review:**

The paper deals with how to efficiently compute how-provenance annotations for SPARQL query results as a means of providing an explanation about the inclusion of triples in a query result. Such result annotations are conceptually polynomial expressions that encode the sources and the data transformations required to obtain a particular query answer and are useful for applications that need to conduct trust assessments of their application data.  The expressions builds on spm-provenance semirings.

The proposed approach NPCS, a Native Provenance Computation approach for SPARQL queries which uses a novel query rewriting technique to integrate provenance expressions into queries using reification. An initial query rewriting is proposed based groupings (Unions) and concatenations (ANDs). Then, a further optimized query rewriting called a "sum query" is proposed to eliminate repeated aggregations in the expressions.  One advantage of using a query rewriting technique is that such techniques are amenable to easy integration in existing SPARQL query engines using reification schemes. In-fact, the authors claim that their proposal is the first 100% SPARQL-based solution for how-provenance. The NPCS supports both monotonic and non-monotonic queries whereas the latter are not supported by many SPARQL provenance query engines.
Sound and completeness proofs of query rewriting are presented.

Evaluation is conducted by comparing performance against another provenance annotation technique called SPARQLProv that is also based on query rewriting. The evaluation objective was to compare the quality of the rewritings produced by both approaches by executing them on two SPARQL query processing platforms: Stardog and GraphDB. The workload was a set of queries selected from the WatDiv benchmark and some queries from the WikiData benchmark. Results show that NPCS outperforms SPARQLProv by a factor of 25.

+ paper is well motivated and written.
+ evaluation and discussion of results about comparative performance against other approaches are insightful
+ the proposed technique seems to significantly outperform alternatives and appears to scale to billions of triples
+ The source code, results and the components to recreate the experimental setup are publicly available on github.
+ Interesting future work discussion

- a few minor editorial errors
a common error pattern is using commas unnecessarily which makes parsing some sentences challenging.
the word minus is written as monus somewhere

**Questions:**

None

-------------------------------------------------------------------------------------------------------------------------------
I appreciate the authors' detailed responses to all the questions that have been raised.

**Reviewer Confidence:**

3: The reviewer is confident but not certain that the evaluation is correct

**Scope:**

4: The work is relevant to the Web and to the track, and is of broad interest to the community

---

### Official Review · Reviewer_E3rV · 2023-12-01

**Novelty:** 7
**Technical Quality:** 7

**Review:**

The authors discuss NCPS, a native provenance computation approach for SPARQL queries. It uses a rewriting mechanism to annotate the query results with their how-provenance. NCPS is evaluated using a synthetic dataset (WatDiv benchmark) and a real-world dataset (WDBench benchmark) on GraphDB and Stardog. The results indicate that NCPS query execution times are much lower than the other system it is compared with (SPARQLProv).

Strengths
1) NCPS is the first native approach for how-provenance.
2) Evaluation is performed on large datasets with several queries (especially using WDBench).
3) This work is relevant to the conference and the track.

Weaknesses
1) NCPS is compared with only one other existing approach.
2) An appendix having a few query rewriting examples and examples showing the usage of rules from Sections 4.2, 4.3 will make the paper more approachable for a wider audience.

Typos
1) Line 138, monus => minus?

**Questions:**

1) Why is NCPS compared with only one system? Comparing with TripleProv along with SPARQLProv would have been good. Why was only one system considered?
2) What are the non-native aspects of the other state-of-the-art approaches when compared with NPCS?
3) Would all the SPARQL queries fit into the rules defined in Sections 4.2, 4.3?

**Reviewer Confidence:**

3: The reviewer is confident but not certain that the evaluation is correct

**Scope:**

4: The work is relevant to the Web and to the track, and is of broad interest to the community

---

### Decision · Program_Chairs · 2024-01-22

**Decision:**

Accept (Oral)

**Comment:**

This article introduces a query rewriting approach to determine the how-provenance of SPARQL queries.

 All reviewers agree that this is a valuable contribution for the Web Conference, and deserves to be accepted.
 We recommend the authors to incorporate the comments and clarifications that arose during the discussions, especially those regarding the formalizations and what the scientific contribution is.